# Utility of Genetic Testing from the Perspective of Parents/Caregivers: A Scoping Review

**DOI:** 10.3390/children8040259

**Published:** 2021-03-27

**Authors:** Robin Z. Hayeems, Stephanie Luca, Daniel Assamad, Ayushi Bhatt, Wendy J. Ungar

**Affiliations:** 1Child Health Evaluative Sciences, The Hospital for Sick Children, Toronto, ON M5G 1X8, Canada; stephanie.luca@sickkids.ca (S.L.); daniel.abdassamad@sickkids.ca (D.A.); abhatt82@uwo.ca (A.B.); wendy.ungar@sickkids.ca (W.J.U.); 2Institute of Health Policy Management and Evaluation, The University of Toronto, Toronto, ON M5T 3M6, Canada; 3Schulich School of Medicine and Dentistry, Western University, London, ON N6A 5C1, Canada

**Keywords:** personal and perceived utility, genetic testing, pediatrics, parent/caregiver perspective, patient reported outcome measure

## Abstract

In genomics, perceived and personal utility have been proposed as constructs of value that include the subjective meanings and uses of genetic testing. Precisely what constitutes these constructs of utility and how they vary by stakeholder perspective remains unresolved. To advance methods for measuring the value of genetic testing in child health, we conducted a scoping review of the literature to characterize utility from the perspective of parents/caregivers. Peer reviewed literature that included empiric findings from parents/caregivers who received genetic test results for an index child and was written in English from 2016–2020 was included. Identified concepts of utility were coded according to Kohler’s construct of personal utility. Of 2142 abstracts screened, 33 met inclusion criteria. Studies reflected a range of genetic test types; the majority of testing was pursued for children with developmental or neurodevelopmental concerns. Coding resulted in 15 elements of utility that mapped to Kohler’s four domains of personal utility (affective, cognitive, behavioural and social) and one additional medical management domain. An adapted construct of utility for parents/caregivers may enable specific and standardized strategies for researchers to use to generate evidence of the post-test value of genetic testing. In turn, this will contribute to emerging methods for health technology assessment and policy decision making for genomics in child health.

## 1. Introduction

Anticipated benefits of existing and emerging genome diagnostics include definitive diagnosis and risk identification, enabling early intervention and a precision medicine approach to management that can potentially reduce morbidity and improve quality of life [1,2]. In addition to identifying genetic variants causally associated with a given indication for testing, genome-wide technologies have the capacity to identify variants associated with pre-symptomatic health risks and pharmaceutical sensitivities [2,3,4]. As such, applications of genome-wide technologies may be particularly beneficial in children since early diagnosis, risk identification and tailored intervention can result in lifelong benefit. Moreover, diagnosis and risk information generated for an index child may have significant health and non-health related implications for family members [5].

Defining and measuring the notion of benefit in the context of genome diagnostics requires the inclusion of metrics that extend beyond laboratory-based performance and incorporate patients’ perspectives [6,7,8,9]. This requirement aligns with a broader transformation in clinical research that has prioritized ascertaining patient and family experiences [10]. While a wide range of patient-reported outcome measures have been used in the genomics literature to reflect on patients’ preferences, experiences, and psychosocial responses to genetic testing [11], patient-oriented utility (i.e., personal or perceived utility) has emerged as a core construct that unifies many of these elements and reflects on the subjective informational value of genetic testing. Personal and perceived utility have been described as constructs of value that include the subjective health and non-health-related meanings and uses of genetic testing and/or a genetic diagnosis, such as increasing knowledge and feelings of control, and optimizing care for individuals and family members in the present and for the future. Some scholars assert that patient-oriented utility (be it ‘personal’ or ‘perceived’) is inclusive of unfavourable effects of genetic testing (i.e., risks related to privacy, discrimination, distress) [12,13,14,15], while others suggest unfavourable effects do not reflect this construct [16,17].

Most recently, Kohler et al. [16] conducted a systematic review of the literature and characterized the concept of personal utility as inclusive of affective, cognitive, behavioural, and social domains. Using a Delphi approach, Kohler et al. [17] established a list of 24 items that represent 14 elements of personal utility, grouped into four domains. Kohler’s construct emphasizes personal utility as a non-health related outcome measure, while others have characterized the value of genetic testing from patients’ perspectives more broadly to include health and clinical-management related impacts. Scheuner et al., [18] for example, engaged patient, clinician, researcher, administrator, and policy maker stakeholders in a modified Delphi process and identified health and medical management as important domains in characterizing the impact of genetic testing. Additional domains identified to be important by this panel included reproductive, diagnostic/prognostic, patient behavioural, patient psychosocial, and family-related. Similarly, from our earlier work with parents of children who received genetic testing, we learned that parents characterize the value of genetic testing as intrinsic in terms of the test’s ability to provide a much sought-after answer for their child’s condition, and as instrumental in terms of its ability to guide care, access health and social services, reduce stigma, make family planning decisions, and engage in proactive prevention strategies [15,19,20]. Finally, Bunnik [12] differentiates perceived and personal utility, emphasizing that personal utility is only achieved when (valid) genetic information is useful (i.e., instrumental) in guiding actions of a personal nature.

What constitutes patient-oriented utility is complex and multi-dimensional. Arguably, how the construct is defined also depends on the clinical context and patient/respondent population to whom it is offered [12,13]. In the pediatric context, generally speaking, measuring patient-reported outcomes is challenged by the fact that children are not typically cognitively capable of making assessments that require abstract thinking and proxy measures (whereby parents respond for their child) do not provide an authentic assessment of the child’s perspective [21,22,23]. In genomics, this measurement challenge is compounded by the fact that an intervention like a genetic test, when performed on an index child, has health and non-health related impacts that extend beyond the child, to parents and other family members. Even if a measurement tool could capture a child’s authentic perspective, it would likely provide a limited understanding of the impact of genetic testing on the family unit overall. As such, although parents/caregivers cannot serve as perfect proxies for their child’s views, they are well-positioned to provide a personal account of the utility of a genetic test for themselves and for the family unit, inclusive of the child. To provide an overview of the literature on utility from the parent/caregiver perspective and to inform the development of emerging value frameworks and health technology assessment methods for precision medicine, [8] we conducted a scoping review of the literature. Building on Kohler’s work [16,17], we focused specifically on *parental/caregiver utility* associated with receiving genetic test results.

## 2. Materials and Methods

### 2.1. Search Strategy

This study complies with the “Preferred Reporting Items for Systematic reviews and Meta-Analysis extension for Scoping Reviews (PRISMA-ScR)” guidelines [24]. Papers that have been central to understanding the construct of patient-oriented utility in the genetics literature [12,14,15,16,17,18,19,20] were used to generate a preliminary list of key words. Search terms included: genetic testing OR genetic counseling AND patient reported outcome OR personal utility OR perceived utility OR personal value OR patient/caregiver preference. The full list of search terms is provided in Appendix A. As described, patient-oriented utility can include health and non-health related outcomes of genetic testing that have either instrumental or intrinsic value to patients and families [12,13,14,15,16,17]. To ensure sufficient coverage of this construct after an initial search, this list was expanded and a full search strategy was developed in consultation with a medical librarian. The SCOPUS, MEDLINE, EMBASE and Web of Science databases were searched for English language articles published between 1 January 2016 and 31 December 2020; to capture the period of time that directly follows Kohler’s critical work in this area [16,17], enabling us to build on their findings.

### 2.2. Study Selection

Search results from all four databases were imported into an EndNote (X7) library. Duplicate articles were identified and removed using the Bramer method [25]. Conference publication titles were removed at the outset where possible. The abstracts of all remaining studies were screened by two authors according to the following inclusion criteria: (i) English language, published between 1 January 2016 and 31 December 2020, contained empirical evidence, peer-reviewed, (ii) study collected data after genetic test results were reported to patients/families, (iii) study design included parent/caregiver reported outcomes, experiences, preferences, or values related to results received from genetic testing (i.e., actual results, not hypothetical scenarios). The following were excluded: conference materials, editorials, consensus statements, practice guidelines, meta-analysis, systematic reviews, opinion or workgroup papers, case reports/series, discrete choice experiments, studies primarily related to diagnostic yield or gene discovery, hypothetical genetic testing, cost effectiveness, or outcome measure development or validation. If it was unclear from the abstract whether or not a study was eligible, it was included for full text review. Eligible studies published before 1 January 2016 were excluded since we anchored our approach to Kohler’s construct of personal utility [16]. Full text review of remaining articles (*n* = 138) was conducted by two independent reviewers and followed the same inclusion criteria as above. Any differences in assessment of eligibility between the two reviewers were resolved by discussion and a third reviewer was consulted where necessary. The percent agreement between raters with regard to eligibility was determined.

### 2.3. Data Extraction and Synthesis

Two authors extracted the following characteristics from each study: PubMed ID, authors, publication year, study title, country of data collection, respondent type, International Classification of Diseases (ICD-11) disease category [26] for index case, genetic test type pursued, sample size, study design, utility-related outcomes, and utility related themes. Where more than one sample was described in an article, the larger sample was identified as the primary sample and the smaller sample was identified as the secondary sample. For example, some studies reported survey findings from the full sample as well as qualitative findings ascertained from a sub-sample of survey participants. Utility referred to parent/caregiver reported outcomes, experiences, preferences, or values related to the information available or received from genetic testing.

Organizing and synthesizing extracted outcomes and themes followed a multi-step process. First, outcomes and themes were extracted from the final eligible pool of studies. These outcomes and themes were assigned to domains of personal utility, as described by Kohler [17] (i.e., affective, cognitive, behavioral, social). An ‘other’ category was included to capture outcomes and themes that did not fall into Kohler’s domains of utility. The two authors who completed full text extractions reviewed domain assignments and resolved discrepancies that emerged through discussion. This mapping exercise endorsed the four domains of personal utility identified by Kohler et al., and identified a medical management domain similar to that identified by Scheuner’s value framework and related empiric work [12,13,14,15,18]. For each Kohler domain, outcomes and themes were mapped to pre-established elements and element concepts. Where our extracted outcomes and themes aligned with Kohler’s element concepts, we retained these concepts. Where new concepts emerged or where concepts warranted re-characterization to suit the parent/caregiver perspective, modifications were made. For the medical management domain, new elements and element concepts were defined [12,13,14,15,18].

### 2.4. Quality Appraisal

A quality appraisal of all studies was conducted. The QualSyst quality assessment criteria for qualitative, quantitative, and mixed methods studies were used (Appendix A: Quality appraisal scores) [27]. For the quantitative studies, 14 items (e.g., assessing study objective, methodology, analysis, conclusions) were scored depending on the degree to which the specific quality criteria were met (“criteria met” = 2, “criteria partially met” = 1, “criteria not met” = 0). Items not applicable to a particular study design were marked “n/a” and were excluded from the calculation of the summary score. A score was calculated for each article by summing the scores obtained across relevant items and dividing by the total possible score. Scores for the qualitative studies were calculated in a similar fashion based on the scoring of ten items. Since assigning “n/a” was not permitted for qualitative studies [27], the total possible score for each qualitative study was 20. For mixed methods studies, an additional five criteria from the Mixed Methods Appraisal Tool (MMAT) [28], assessing the integration of qualitative and quantitative methods and conclusions Appendix A, were rated on the same 0 to 2 scale. Final scores for mixed methods studies were determined by adding the scores from the qualitative, quantitative, and mixed methods appraisal tools, divided by the total possible score [28]. Quality score averages and ranges were calculated Table 1.

## 3. Results

### 3.1. Study Selection, Characteristics and Quality

Our comprehensive search yielded 4499 abstracts. Duplicates were removed and the remaining 2142 abstracts were screened for inclusion. After removing 2004 conference or literature review abstracts that did not meet inclusion criteria, 138 full-text papers were reviewed. Of these, 105 were excluded based on lack of empirical results, or study designs that focused on hypothetical response to genetic testing, pre-test perceptions of genetic testing, and/or non-parent/caregiver reported outcomes. Thirty-three [15,29,30,31,32,33,34,35,36,37,38,39,40,41,42,43,44,45,46,47,48,49,50,51,52,53,54,55,56,57,58,59,60] met all inclusion criteria Figure 1. All full texts were reviewed for inclusion by two independent reviewers, with an inter-rater agreement of 83.4%. Absence of consistency between raters pertained to raters’ oversight of study details related to inclusion criteria.

The majority (55%) of the studies were conducted in the United States. The remaining studies were conducted in Europe (18%), Australia (15%), Canada (9%), and New Zealand (3%). Of the 33, 20 (61%) were qualitative, nine (27%) were quantitative, and four (12%) were mixed methods. In decreasing order, the most common genetic tests on which study participants reflected were chromosome microarray (*n* = 16, 48.5%), whole exome sequencing (*n* = 14, 42.4%), multi-gene panel (*n* = 6, 18.2%), and karyotype (*n* = 5, 15.2%). The 33 studies included a total of 3606 primary sample participants and 93 secondary sample participants. Respondent type included: parents/caregivers of minor patient (73%), prospective parents (i.e., prenatal testing context; 16%), caregivers of adult patient (8%), and other (3%). The most common clinical conditions under study were developmental anomalies (61%), behavioural or neurodevelopmental disorders (39%), prenatal or pre-implementation genetic testing (27%), and diseases of the nervous system (21%) (Table 1). The complete list of included studies is provided in Appendix A: Scoping review studies and assigned study number.

With respect to quality, quantitative studies reviewed scored the highest, with an average score of 82.9%, compared to qualitative (81.0%) and mixed methods (80.8%) studies. Almost all qualitative studies were given a perfect rating for study design; the reflexivity item (i.e., extent to which authors consider their own characteristics or worldview during data collection and analysis) scored the lowest. All quantitative studies had clear study objectives and design; the lowest scoring item related to controlling for confounding. Two of the four mixed methods studies did not address inconsistencies between qualitative and quantitative findings well. No studies were excluded based on their quality appraisal score Appendix A.

### 3.2. Utility from Parents’/Caregivers’ Perspectives

The outcomes and themes identified in 33 studies that were specific to the receipt of results from genetic testing from parents’/caregivers’ perspectives aligned with Kohler’s four domains and 14 elements of utility. The outcomes and themes characterized as element concepts associated with each element of each domain are displayed in Table 2. The element specific to “feelings of responsibility” (affective domain) was most commonly reported (72.7% of studies). The element specific to “knowledge of condition” (cognitive domain) was reported in 69.7% of studies and the elements of “reproductive autonomy” (behavioural domain) and “change in social support” (social domain) were reported in 60.9% of studies. In addition, we identified a management domain which included an element related to clinician-directed management activities and element concepts related to initiation of or alteration to medication use, ongoing diagnostic testing, surveillance and monitoring, and sub-specialist or community referrals. Clinician directed activities applied to both probands and family members. This domain of utility was identified in 48.5% of studies reviewed.

While Kohler’s four domains of personal utility were identified in the studies reviewed, our findings extend the definitions of some of Kohler’s domain elements to include aspects of utility that were germane to parents/caregivers. While these aspects may be implicit in the definitions of the existing elements in Kohler’s framework, our review prompted us to explicitly name these aspects as important parts of the construct of utility when thinking about it from parent/caregiver perspectives.

In the affective domain, where outcomes of genetic testing refer to an individual’s emotional state, [17] the studies reviewed suggest the addition of concepts to three affective elements. For ‘enhanced coping’, studies reviewed include peace of mind and relief and for ‘mental preparation’, studies reviewed include the concept of hope. For ‘feelings of responsibility,’ studies reviewed include the concepts of guilt, worry and sense of competence. In the cognitive domain, where outcomes refer to gains in knowledge or information from genetic testing, studies reviewed suggest the addition of concepts to each of the four elements. For ‘value of information’, studies reviewed emphasize the concept of closure and for ‘knowledge of condition’, studies include explicit knowledge related to diagnosis, prognosis, and recurrence risks. For ‘curiosity’, studies reviewed emphasize a need to satisfy curiosity and seek information. For ‘self-knowledge’, studies reviewed emphasize the importance of understanding the child (i.e., not oneself) with the medical condition.

In the behavioural domain, where outcomes represent practical uses of genomic information, studies reviewed suggest the addition of concepts to each of the three existing elements of utility. Regarding ‘ability for future planning’, studies reviewed include impact on career, finances and lifestyle for parents. Regarding ‘reproductive autonomy’, studies reviewed extend this element to include informing planning of future pregnancies not only for the index case but also for other family members. Regarding ‘communication’, while communication with family members is inclusive of all members, studies reviewed describe communication with the child as a distinct concept. Finally, in the social domain, where outcomes of genetic testing involve changes in social support or status on individual, familial, and societal levels, the only additional concept identified in the studies reviewed related to the ‘concern over discrimination’ element. Similar to Kohler’s systematic review [16], studies reviewed here indicated that privacy concerns warrant inclusion in this element.

## 4. Discussion

Adapting the construct of personal utility derived from Kohler’s systematic review [16] and Delphi’s study [17] for parents/caregivers, we identified 15 distinct elements of utility spanning five non-overlapping domains. While two elements (i.e., self-knowledge and feeling good for helping others) were identified in only three studies reviewed and one (i.e., spiritual well-being) was identified in only one, all other elements were identified in five or more independent studies, indicating that these elements comprise a good representation of the construct of utility, as perceived by parents/caregivers. Parents of children with developmental or neurodevelopmental indications for genetic testing represented the most common perspective captured. While most included studies focused on these clinical indications, a range of genetic test types (i.e., microarray, single gene, gene panel, exome sequencing) were considered by parent/caregiver respondents, including some prospective parents. As such, findings herein are not limited to the context of exome/genome sequencing but apply to a range of diagnostic genetic testing strategies.

Our findings map neatly onto four domains and 14 elements of personal utility identified by Kohler [16,17]. Within the 14 elements, our findings suggest that important modifications are warranted to characterize the construct of utility from parents’/caregivers’ perspectives. These include (i) the addition of the concepts of hope, guilt, worry, competence, peace of mind and relief to the affective domain, (ii) the addition of the concepts of closure, information seeking, knowledge related to diagnosis, prognosis, and recurrence risks and explicit attention to understanding the child to the cognitive domain, (iii) explicit differentiation of behavioural impacts of test results on parents, children, and other family members in the behavioural domain, and (iv) the inclusion of the concept of privacy in the social domain. While adding concepts related to peace of mind and privacy represent an adaptation to the elements defined in Kohler’s Delphi study [17], these concepts were identified in the systematic review [16] they conducted to inform the Delphi process.

A significant difference between Kohler’s characterization of personal utility and that defined by the studies reviewed herein is the addition of the medical management domain. In related literature about genetic test evaluation, medical management implications are more often described as a component of the clinical utility construct. While many have argued that medical management implications constitute health-related outcomes and fit within the construct of clinical utility, others consider medical management as an example of an indirect or intermediate non-health related outcome, since a medical management decision, unto itself, does not comprise a health outcome [12,13,61,62,63,64,65,66,67,68,69]. While the distinction between health and non-health related outcomes remains important to the task of defining and adjudicating the value of genetic testing [67,68,69], it is plausible that both health and non-health related outcomes can be perceived and experienced from both clinical and personal perspectives. There is justification for medical management implications, as an impact of genetic testing that sits at the boundary of health and non-health related outcomes, to be included in both personal and clinical utility constructs. Findings from the present review suggest that medical management implications are an important aspect of utility from parent/caregiver perspectives and warrant inclusion in the definition of this construct.

Several patient-reported outcome measures have been validated for use in genetics, many of which include elements of the construct of personal utility defined by Kohler et al. [16,17] as well as the modified construct defined herein. For example, McAllister et al. [70] and Grant et al. [71] developed and validated the Genetic Counseling Outcome Scale (GCOS-24) and the Genomics Outcomes Scale (GOS), respectively, to capture the impact of genetic counselling services on patient empowerment. While the dimensions of the GCOS (i.e., cognitive, decisional and behavioural control, emotional regulation, and hope) reflect on non-health-related benefits of genetic counselling and overlap with dimensions identified by Kohler and by our review, the measures are not specific to the impact of genetic testing, rather they are focused on the impact of genetic counselling services. Using a different theoretical underpinning than the GCOS and GOS, McConkie-Rosell et al. developed the Genome Empowerment Scale (GEmS) [72]. While this tool was designed to understand parents’ perspectives on the process of empowerment in the context of genetic testing, it is administered prior to testing and does not reflect on the actual impact of test results. Yusuf et al.’s measure of utility of biomarker testing (Perceived Utility of Biological (PUB) Testing) also focuses on the anticipated pre-test value of testing rather than the outcomes of testing [73]. The Feelings About genomiC Testing Results (FACToR) scale measures the post-test psychosocial impact of genomic findings [74] but includes only the affective and cognitive domains of the personal utility construct defined by Kohler et al. [16,17] Findings from the studies reviewed here suggest that the construct of utility, oriented around the post-test impact of genetic testing, from parents’/caregivers’ perspectives, can be differentiated from constructs contained in existing patient reported outcome measures in genomic medicine.

While findings from this work provide a more detailed characterization of the construct of utility from parents’/caregivers’ perspectives, we acknowledge specific limitations. First, study inclusion criteria were challenging to establish and apply given the absence of an agreed upon definition of the patient-oriented utility in the field of genomics to date. In part, however, the purpose of the review was to contribute to efforts to define this construct for parents/caregivers. Second, we excluded non-English studies, potentially leading to our omission of concepts relevant to utility for parents/caregivers. We also excluded discrete choice experiments, cost effectiveness studies, and outcome measure development or validation studies. While discrete choice experiments provide preference data and use choice sets that are typically derived from the lived experience of the target population, the data generated are technically hypothetical. Cost effectiveness and measurement development/validation studies may reflect patient-oriented utility but the intent of these studies is distinct from our focus on further characterizing the construct of utility from parents’/caregivers’ perspectives. Third, while we view our use of Kohler’s personal utility construct as a strength and as a reference point from which to tailor the construct of utility to parents/caregivers, it may have acted to constrain our thinking about the elements and concepts embedded in this construct, and may have limited the scope of the literature reviewed given that 2016 was our starting point. The exclusion of studies published in 2021 is a further limitation. Finally, we included studies of variable quality, potentially representing a reporting bias.

Patient-oriented utility has emerged in the genomics literature as a critical measure of value. While it is generally agreed that personal and perceived utility reflect on the subjective meaning an individual ascribes to genetic testing, consensus has not emerged on the precise definition of these and related constructs. For example, whether or not elements of these constructs are distinct from or overlapping with elements of clinical utility or with elements of psychosocial well-being remains contested [9,16,63]. In addition, some orient to personal and perceived utility as measures of anticipated (i.e., pre-test) value [72,73], some as a measure of actual (i.e., post-test) value [15,20,74,75], and some as a composite of both [16,17,70,71]. Finally, whether any one of these constructs can be applied to a range of clinical genetics settings and respondent types or whether tailored approaches are required to achieve face validity in a range of settings requires further consideration. Building on Kohler’s evidence-informed approach to define specific elements of personal utility [16,17], we offer a modified version of utility that attends to the parent/caregiver perspective.

The field of genomics presents a multitude of challenges for assessing the value of emerging technologies through formal health technology assessment [8,67,68,69] and for health technology assessment in child health in particular [76]. The construct of patient-oriented utility is gaining prominence in aspects of health technology assessment related to assessing patient preferences as well as the consideration of ethical, legal and social implications [77]. Refining the construct of utility to suit the parent/caregiver perspective presented in this review and developing strategies in the future to operationalize the measurement of parent/caregiver utility will contribute to methodological innovation in this field and a richer evidence base from which informed policy decisions about technology adoption can be made.

## Figures and Tables

**Figure 1 children-08-00259-f001:**
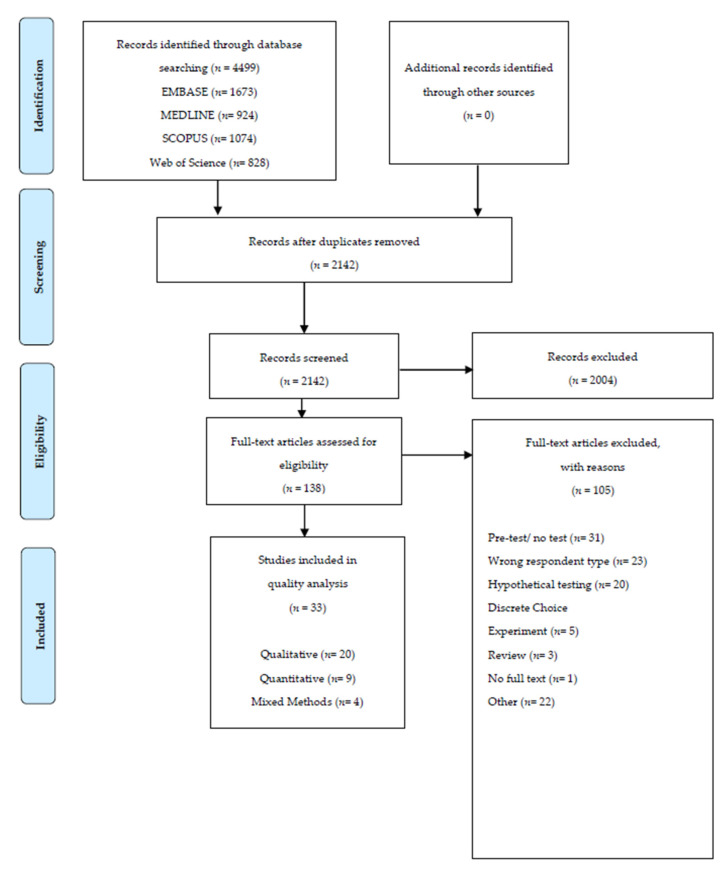
PRISMA flow diagram.

**Table 1 children-08-00259-t001:** Study characteristics (*n* = 33).

Studies Included	Count	%
2020	8	24.2
2019	3	9.0
2018	7	21.2
2017	3	9.0
2016	12	36.4
**Characteristic**	**Count**	
**Country of Data Collection**		
USA	18	54.5
Europe	6	18.2
Australia	5	15.2
Canada	3	9.1
New Zealand	1	3.0
**Population (Respondent Type) ^a^**		
Parent/Caregiver of minor patient	27	73.0
Prospective Parent	6	16.2
Caregiver of adult patient	3	8.1
Adult sibling caregiver of minor patient	1	2.7
**ICD-11 Disease Category ^a^**		
Developmental anomalies	20	60.6
Mental, behavioural or neurodevelopmental disorders	13	39.4
Diseases of the nervous system	7	21.2
Prenatal or pre-implementation genetic testing ^b^	9	27.3
Neoplasms	2	6.1
Diseases of the blood or blood-forming organs	2	6.1
Diseases of the respiratory system	2	6.1
Other ^c^	5	15.2
Unreported	9	27.3
**Genetic Test Type ^a^**		
Karyotype, FISH, Fragile X, Single gene test	12	36.4
Chromosomal microarray	16	48.5
Multi-gene panel	6	18.2
Whole exome sequencing	14	42.4
Whole genome sequencing	4	12.1
Prenatal ^d^	4	12.1
Newborn screening	2	6.1
Other ^e^	2	6.1
Unreported	1	3.0
**Study Type and Quality Scores**		
Qualitative	20	60.6
Average quality score (Range)	-	81.0 (65.0–95.0)
Quantitative	9	27.3
Average quality score (Range)	-	82.9 (64.0–95.5)
Mixed methods	4	12.1
Average quality score (Range)	-	80.8 (75.0–92.3)
**Total Primary Sample Size**	3606	
**Total Secondary Sample Size**	93	

^a^ Multiple characteristics may apply to a single paper; ^b^ the ICD-11 classification “Pregnancy, childbirth or the puerperium” was relabeled to “Prenatal or pre-implementation genetic testing” for the purpose of this review to more accurately describe the studies in this category; ^c^ other disease category = Endocrine, nutritional or metabolic diseases, diseases of the circulatory system, diseases of the visual system, visibly healthy, recessive and X-linked conditions in offspring; ^d^ prenatal genetic test type = Non-invasive prenatal testing (NIPT), pre-implantation genetic screening; ^e^ other genetic test type = Tumour sequencing or commercial publicly available. Abbreviations: FISH = fluorescence in situ hybridization.

**Table 2 children-08-00259-t002:** Elements of personal utility from the perspective of parents/caregivers.

Domain	Element	Element Concepts *Genetic Test Results…*	Number of Studies per Element	Percentage of Studies (*N* = 33)	Studies
*Medical Management*	*Clinician-directed activities*	*Enable or improve access to health/services (e.g., surveillance, referrals for proband and family members)* *Alter treatment, medication* *Limit/reduce testing, surveillance*	16	48.5%	3–7, 9, 12, 13, 15, 16, 18, 19, 22, 26, 27, 28
Affective	Enhanced coping	Enable one to cope with *child’s* health risks Enable one to feel more in control of self *and child* Enable one to feel more in control of life situation, for self *and child* *Spur feelings related to peace of mind* *Spur feelings related to relief*	14	42.4%	2, 7, 11, 13, 17, 20–23, 26, 28, 30–32
	Mental Preparation	Help one or one’s family mentally prepare for the future Give one a false sense of security *Spur feelings related to hope*	19	57.6%	4, 5, 7–10, 12, 17, 20–22, 24, 26–32
	Feeling of responsibility	Spur feelings of responsibility for child’s health and risks *Spur feelings of guilt or worry* *Spur feelings related to parenting competence*	24	72.7%	3–5, 7–10, 12, 13, 15, 17, 20–24, 26–33
	Improved Spiritual Wellbeing	Enable one to live more fully	1	3.0%	30
Cognitive	Value of Information	Are valuable simply because they provide information Are valuable no matter what the results are *Are valuable because they provide closure*	12	36.4%	9, 11, 14, 16–18, 21, 22, 26, 28, 29, 31
	Knowledge of Condition	Help one understand one’s health condition better *(i.e., diagnosis, prognosis and recurrence risk)*	23	69.7%	2–5, 7–9, 11–15, 17–20, 22, 23, 26, 27, 29–31
	Curiosity	Satisfy one’s curiosity *Prompt information seeking*	8	24.2%	1, 4, 7, 10, 21, 22, 30, 33
	Self-Knowledge	Improve one’s self-knowledge Improve one’s understanding of one’s family *Improve one’s understanding of the child with their condition*	3	9.1%	12, 17, 26
Behavioural	Ability for future planning	Allow one to organize long-term care Motivate one to get one’s affairs in order Inform one’s plans for school or career *Impact parents’ career, finances, or lifestyle*	9	27.3%	4, 5, 8, 9, 12, 18, 26, 27, 31
	Reproductive Autonomy	Inform one’s decisions about having children Can be used for prenatal testing to ensure risk information is provided in current and future pregnancies *Inform future pregnancies for others (e.g., parent, child, relative)*	20	60.6%	2–4, 7–9, 11, 13, 15, 18, 21, 22, 25–30, 32, 33
	Communication	Spur increased communication with one’s family members. *Spur disclosure of results to child*	8	24.2%	1, 4, 5, 9, 17, 22, 26, 30
Social	Concern over discrimination	Make one nervous about discrimination; insurance, employment, *and privacy*	7	21.2%	1, 4, 10, 15, 22, 26, 30
	Feeling good for helping others	Make one feel good for contributing to research Make one feel good for providing knowledge to one’s family	3	9.1%	15, 22, 31
	Change in Social Support	Lead to greater support from one’s friends and family Allow one to take advantage of social programs; advocacy Enable child to access social services	20	60.6%	1, 3, 4, 7–9, 12, 15, 16, 18, 19, 21–23, 26, 30, 31

Notes: Concepts in italics represent a new personal utility domain (i.e., medical management) and modifications to Kohler’s element concepts as per Kohler et al. 2017 [17]. Study numbers correspond to Appendix A.

## Data Availability

All data generated or analysed during this study are included in this published article (and its Appendix A).

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
