# Peer review of "Utility of Genetic Testing from the Perspective of Parents/Caregivers: A Scoping Review"

_children, 2021, doi:10.3390/children8040259_

Round 1
Reviewer 1 Report
Thank you for the opportunity to review this well-written manuscript “Personal utility of genetic testing from the perspective of parents/caregivers: A scoping review”. Overall, the research makes an important contribution to the literature and understanding about the value of genetic testing. There are a number of suggestions that could strengthen the manuscript, outlined below in order of appearance:
Introduction:
- On page 2 the authors state that personal utility is “inclusive of unfavorable effects of genetic testing (i.e. risks related to privacy, discrimination, distress” and cite (among others) Kohler et al., 2017 (reference 13). However, Kohler et al., in their Delphi study discuss that such negative outcomes or harms do not themselves constitute utility. It may be helpful to mention somewhere in the manuscript, that inconsistency remains about whether negative outcomes should be included in a definition of utility.
- At the end of the Introduction or start of Methods please provide a rationale for why a scoping review is suitable for the research question.
Methods:
- The review covers a period of 3.5 years, though at the time of reviewing, the search was conducted over 1.5 years ago. According to the Cochrane review guidelines:
“The published review should be as up to date as possible. The search must be rerun close to publication, if the initial search date is more than 12 months (preferably 6 months) from the intended publication date, and the results screened for potentially eligible studies. Ideally the studies should be fully incorporated. If not, then the potentially eligible studies will need to be reported, at a minimum as a reference under ‘Studies awaiting classification’ or ‘Ongoing studies’.” - Chandler et al. "Methodological standards for the conduct of new Cochrane Intervention Reviews." Sl: Cochrane Collaboration (2013).
Given the speed of advances in genomics, it would be preferable for the review to be updated, or at least report eligible studies as suggested above. For example this article could be of importance to readers: Mollison et al. "Parents’ perceptions of personal utility of exome sequencing results." Genetics in Medicine (2020): 752-757.
The limitations section notes the search date; if resource limitations prohibit the review from being updated, this should be stated. - It would be helpful to include details about the rationale for included studies. For example, why were some study types such as discrete choice experiments not included?
- How many authors reviewed the abstracts? Please clarify if only conference and review abstracts were excluded at the abstract screening stage, or if other inclusion criterion were assessed at the abstract stage.
- Please clarify if both independent reviewers conducted the full text review on all articles (i.e. two reviewers read every full text), or if only a portion were read by two reviewers.
- How many people performed the data extraction and how was reliability assessed?
Results:
- The study characteristics are presented in Table 1 but this section does not include citations of the included studies. Please reference the supplemental material somewhere in this section.
- The paragraph describing the quality appraisal is comprehensive, and perhaps contains unnecessary detail. Some of this could be reduced, or at least re-written to use consistent language as in the tool to describe quality. For example, rather than “Of the two mixed methods studies, neither addressed divergences and inconsistencies between qualitative and quantitative findings well” use the descriptive language in the tool that determined the rating of 0 or 1 for this category.
The Discussion section is comprehensive and no further recommendations are indicated.
Reviewer 2 Report
This manuscript describes a very well executed study and an important topic: it elaborates on what constitutes parental perceived utility in genetic testing in child health by conducting a scoping review of the literature. I do question however if that was the aim of the study, since they present the work as if they looked at personal utility, which in my opinion (and in reflection on the work of Kohler et al. and Bunnik et al.) is generally considered a different (more narrow) definition.
In general, the background literature is extensively described, showing the complexity of measuring and describing utility of genetic testing and the relevance of context in defining a construct as personal/perceived utility. Therefore it is clear that specifically looking at personal/perceived utility from the parents perspective would add to current knowledge.
I have three main questions that, in my opinion should be addressed:
- What is the general relevance of the construct of personal(?) utility? How do the authors foresee that this contributes to emerging methods for HTA? What stakeholders would use this construct and how? (and this leads to my following question);
- How is this construct of personal (?) utility used in addition to e.g. measuring clinical utility? By choosing to include aspects of clinical impact, there seems to be overlap between the two, which does not coincide with how Kohler et al. recently characterized personal utility. Doesn’t that make the construct as presented here better reflect (parental) perceived utility i.s.o. personal utility?
- How do the authors see the construct of personal utility in relation to terms as “utility of personal genomic information (ref 15)”, “patients’ perceived utility” (ref 16), “parents perspectives” (ref 19)? They don’t seem to make a distinction, which is also reflected in the terms included in the search. Furthermore, by adding concepts beyond actions (feelings etc.) they also seem to deviate from the existing definition of personal utility.
Some more specific suggestions/comments/questions:
Abstract:
Line 12-13: “What constitutes the construct of personal utility and whether this varies by perspective, remains unresolved.”
I don’t think it is unclear that a construct of personal utility varies by perspective: it is inherently subjective and personal and therefore aspects that are relevant to one could be irrelevant to another.
Line 12-14: “To advance methods for measuring the value of genetic testing in child health…”
How do the authors foresee that personal utility will be measured and by whom will decision based on personal utility be made? Please make more explicit why it is important that we measure personal utility.
Introduction:
Line 46-48: “…personal utility has emerged as a core construct that unifies many of these elements and reflects on the informational value of genetic testing for patients and family members.”
I think the word “subjective” should be added before “informational”, because I think all evaluation of genetic tests (also e.g. clinical validity and utility) is reflecting on the value of the information that a test provides. It is the subjective nature that makes it of specific personal value.
Line 49: While the abstract suggests personal utility is defined by subjective meanings and uses (line 10-11: “…that includes the subjective health….”), in the introduction the authors state that the construct is not limited to these subjective meanings and uses. This is an example of what makes it unclear what precisely the authors feel personal utility would entail and how it relates to the concept of clinical utility.
Line 60-66: “Kohler’s construct emphasizes personal utility as a non-health related outcome measure, while others have characterized the value of genetic testing from patients’ perspectives more broadly to include health and clinical-management related impacts. Scheuner et al, [18] for example, engaged patient, clinician, researcher, administrator, and policy maker stakeholders in an expert-panel, modified Delphi process and identified health and medical management as important domains in charactersing the impact of genetic testing.”
The authors here acknowledge that the construct as described by Kohler only includes non-health related outcome measures. It is however unclear why the authors differentiate from this definition, especially since Sheuner et al. did not describe personal utility, but value/impact of genetic testing in a broader sense. (and a minor spelling mistake in the word characterizing)
Line 97: The authors state to focus on “parental personal utility” in the introduction, but also included studies describing views of prospective parents and an adult sibling of a minor patient. Perhaps more correctly describe as personal utility of (prospective) parents or caregivers (as also reflected in the title of the manuscript)?
Methods:
The authors provide a very elaborate quality assessment and assurance for the methodology used.
Line 108: SupplemEntary is misspelled
Line 203: An inter-rater agreement of almost 84% is described for the review for inclusion. It is however unclear how disagreement was resolved (are these studies included or not?) and what the arguments for disagreement were. Perhaps examples can be given?
Discussion:
Line 345-347: “While the distinction between health and non-health related outcomes remains important to the task of defining and adjudicating the value of genetic testing, it is plausible that both health and non-health related outcomes can be perceived and experienced from both clinical and personal perspectives.”
Although definitely of value from a personal perspective, this does not make health related outcomes part of personal utility per se.
Round 2
Reviewer 2 Report
Previous comments are very well addressed by the authors and I think the manuscript has improved by the edits made since the last version (including the update of the recent literature). The title now also reflects the content.
Although the word "personal" was ommitted rightfully in most cases, sometimes it should however be used, e.g. when referring to Kohler's work: please check this throughout the paper. I think for example in line 678 (and similarly line 692) it would be good to leave it, because the 14 elements reflect aspects of personal utility.
I spotted one spelling mistake in line 578: inClude
